# Fractional Correspondence Framework in Detection Transformer

Masoumeh Zareapoor
Shanghai Jiao Tong University

Pourya Shamsolmoali[*][†]
East China Normal University
Queen's University Belfast

Huiyu Zhou
University of Leicester

Yue Lu[†]
East China Normal University

Salvador García
University of Granada

## ABSTRACT

The Detection Transformer (DETR), by incorporating the Hungarian algorithm, has significantly simplified the matching process in object detection tasks. This algorithm facilitates optimal one-to-one matching of predicted bounding boxes to ground-truth annotations during training. While effective, this strict matching process does not inherently account for the varying densities and distributions of objects, leading to suboptimal correspondences such as failing to handle multiple detections of the same object or missing small objects. To address this, we propose the Regularized Transport Plan (RTP). RTP introduces a flexible matching strategy that captures the cost of aligning predictions with ground truths to find the most accurate correspondences between these sets. By utilizing the differentiable Sinkhorn algorithm, RTP allows for soft, fractional matching rather than strict one-to-one assignments. This approach enhances the model's capability to manage varying object densities and distributions effectively. Our extensive evaluations on the MS-COCO and VOC benchmarks demonstrate the effectiveness of our approach. RTP-DETR, surpassing the performance of the Deform-DETR and the recently introduced DINO-DETR, achieving absolute gains in mAP of **+3.8%** and **+1.7%**, respectively.

## CCS CONCEPTS

• **Computing methodologies → Object detection**.

## KEYWORDS

Object detection, Matching problem, Sinkhorn algorithm.

### ACM Reference Format:

Masoumeh Zareapoor, Pourya Shamsolmoali, Huiyu Zhou, Yue Lu, and Salvador García. 2024. Fractional Correspondence Framework in Detection Transformer. In *Proceedings of the 32nd ACM International Conference on Multimedia (MM'24), October 28-November 1, 2024, Melbourne, Australia.* ACM, New York, NY, USA, 9 pages. https://doi.org/XXXXXXX.XXXXXXX

## 1 INTRODUCTION

Object detection aims to identify and localize objects within images across various categories. The advent of deep learning has significantly enhanced object detection, enabling models to achieve high accuracy and robustness across complex settings [41]. Central to the efficacy of these models is the matching process- how predictions are accurately aligned with ground-truth objects. Accurately

---
[*]Contribute equally with the first author.

[†]Correspondence to (yrui, ylu)@cee.ecnu.edu.cn.

---
*ACM MM, 2024, Melbourne, Australia*
2024. ACM ISBN 978-1-4503-XXXX-X/18/06
https://doi.org/XXXXXXX.XXXXXXX

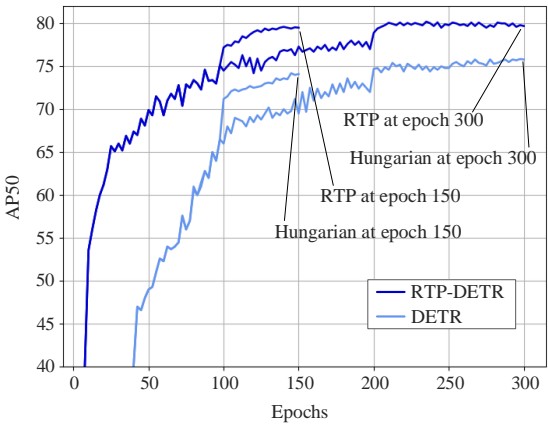

**Figure 1: Learning curves (AP$_{50}$) for DETR and RTP-DETR using a ResNet-50 Backbone, across different training durations on the VOC dataset. Results are shown for training durations of 150 (short) and 300 (long) epochs, with adjustments to the learning rate at the epochs of 100 and 200.**

pairing each predicted object with its ground-truth is a crucial and challenging task [2]. Conventional matching strategies in object detection, such as those used by two-stage detectors (e.g., Faster R-CNN) [27] and one-stage detectors (e.g., YOLO, SSD) [20, 26], rely on predefined anchor boxes and intricate overlap metrics (e.g., IoU) for aligning predictions with ground-truths [8, 41]. Despite their widespread use, these heuristic-based methods limit the model's ability to learn optimal matching from data directly [2, 13, 18], and add complexity due to manual anchor and threshold adjustments.

The Detection Transformer (DETR) [2] emerges as a promising solution, introducing an end-to-end framework that simplifies the object detection pipeline by using the Hungarian matching algorithm [15]. This algorithm provides unique one-to-one correspondences between predicted and ground-truth objects, optimizing the matching cost under the assumption of equal set sizes. In cases, where the two sets do not have the same size, significant preprocessing is required to construct a square cost matrix [13, 16, 40]. DETR addresses this challenge by generating a fixed number of bounding box predictions for each image, with the model learning to classify excess predictions as "no object". However, this way can not fully handle densely packed objects or significantly small objects, as the algorithm's cost function primarily guides these matches without directly considering the overall spatial and class distribution of objects [1, 7, 19, 25, 40]. It also requires a long training schedule to

converge, as illustrated in [7, 42], and Figure 1. Several approaches are proposed to address the prediction-ground truth matching in the DETR, each aiming to navigate the complexities of diverse scenes with greater efficacy. Zhao et al. [40] showed that one-to-one matching does not provide direct supervision for generating predicted objects. They introduced MS-DETR which combines one-to-one and one-to-many strategies, enriching training by considering both individual predicted objects and their context. [39] also argues that the one-to-one assignment often falls short due to the variable distributions and densities of the objects.

Instead of relying on a fixed threshold to discard unlikely pairs, this work computes a marginal probability for each object pair, and provides a more stable basis for object pairing across diverse conditions. Hou et al. [13] improves the matching technique by introducing a salience score to evaluate the relationship between detected and actual objects. This score is designed to ensure an accurate matching, addressing the Hungarian algorithm's limitations in discriminating between closely situated or similar objects. Rank-DETR [25] addresses the mismatch between confidence scores and localization accuracy of predicted bounding boxes. By prioritizing predictions that have more accurate localization, it seeks to improve the quality of matches between detection and actual objects. DETR-like detectors, despite their remarkable performance, rely on the Hungarian algorithm for one-to-one correspondences between detected objects and ground truths, and they integrate additional strategies to handle one-to-many matching scenarios.

Our work goes beyond this traditional method by looking for a reliable way to match sets of predictions with ground-truths of potentially diverse sizes. We preserve the core property of the Hungarian algorithm—minimizing the assignment cost— while extending its capabilities to address fractional assignments and distribution discrepancies efficiently. Our matching technique is based on optimal transport (OT) theory [24], optimizing the transport plan to minimize the cost across the entire distribution rather than focusing on individual matches. Central to our model is the computation of a transport plan, $\Gamma$, where each element $\Gamma_{ij}$ represents the weight (degrees of matching) between predicted object $i$ and ground-truth $j$. This plan derives from an optimization process aimed at reducing the overall cost $C$ between prediction-ground truth pairs. Unlike the Hungarian algorithm, which treats matching as a binary decision, the transport plan assigns weights to each pair of prediction-ground truth, capturing the degree of match. Then, through these weighted assessments, our model finds the best correspondence between prediction and actual objects. OT, by considering the distribution of all predictions and ground-truths, allowing for a more nuanced, probabilistic matching that accurately detects small objects that can be overlooked under a strict one-to-one matching scheme.

**In summary:** ❶ We verify that DETR with the Hungarian algorithm suffers from slow convergence and accurate matching for complex senses. ❷ We propose a regularized transport plan to find the best alignment between predictions and ground-truths, showing how regularization can improve the convergence (see Figure 3). ❸ Our experimental results show that our model outperforms existing DETR-based methods, including Deform-DETR [42], DN-DETR [16], Rank-DETR [25], and other training-efficient variants DINO-DETR [38] and Stable-DETR [19].

## 2 MATCHING FLEXIBILITY

In any matching process, it is essential to establish a matching cost that quantifies the degree of alignment between two sets. Figure 2 compares various matching strategies on an image from the VOC training dataset. The matching cost for each pair of prediction and ground-truth bounding boxes is calculated using the Generalized Intersection over Union (GIoU). In the input image-**(a)** ground-truth objects are color-coded and the prediction boxes are in black. **(b)** displays a cost matrix that measures the matching between predictions and ground-truths, where rows (1-5) represent predicted bounding boxes and columns (A, B) denote ground-truth boxes. Column C is the background cost used in the DETR models for cases of excess predictions over ground-truth objects, guiding the model on which predictions to discard or classify as background. The color intensity is the magnitude of the cost, where a dense color denotes a higher cost (poor matching), and a lighter color indicates a lower cost (better matches). With a background cost set at 0.75, any prediction-ground truth pair exceeding this threshold is considered a match with the background and it is a false positive.

**(c)** shows the RTP with regularization $\epsilon = 0$ and $\kappa_2 = 0$. This allows some ground-truths to receive multiple matches, while others get none. This is effective when the goal is to reduce false positives, ensuring predictions closely align with actual objects. **(d)** illustrates the one-to-one matching by the Hungarian algorithm, ensuring each ground-truth pairs with a single prediction. This is ideal for cases requiring strict correspondence between predictions and ground-truths without duplicates. However, it may not capture the full complexity of object interactions or the presence of multiple objects in close proximity [7, 13, 25, 40]. It only enforces a match based on the lowest cost function but misses complex details due to the constraints of choosing a single best match for each prediction or ground-truth. Moreover, the Hungarian algorithm does not account for the gradual improvement of predictions during training, slowing the model convergence [1, 13, 19, 42]. **(e)** RTP with $\epsilon = 0$ and $\kappa_1 = 0$ demonstrates that each ground-truth is matched with the best prediction, ensuring all actual objects are detected. As a result, a single prediction can match with multiple ground-truths, which is ideal for cases where missing any object is a greater concern than duplicate detections. **(f-h)** present different settings of RTP by adjusting the regularization parameters: (f) sets $\epsilon = 0.05$, $\kappa_1 = 100$, and $\kappa_2 = 0.01$; (g) uses $\epsilon = 0.05$ with both $\kappa_1$ and $\kappa_2$ set to zero; (h) uses $\epsilon = 0.05$, $\kappa_1 = 0.01$, and $\kappa_2 = 100$. We can consider these results optimal when there is a need for a balance between precision and recall. This analysis can highlight the advantages of regularized OT over the Hungarian algorithm, especially to manage discrepancies in the number of predictions and ground-truths, and to provide a nuanced matching technique for complex object detection tasks. Our insights align with the findings in [1, 7, 38, 42].

## 3 RELATED WORK

### 3.1 DETR with different matching frameworks

DETR [2] revolutionized object detection by presenting it as a set prediction problem, using one-to-one matching supervised by the Hungarian algorithm for end-to-end training. Several subsequent works have proposed to address the slow convergence of DETR from different perspectives. [33] stated that the cross-attention

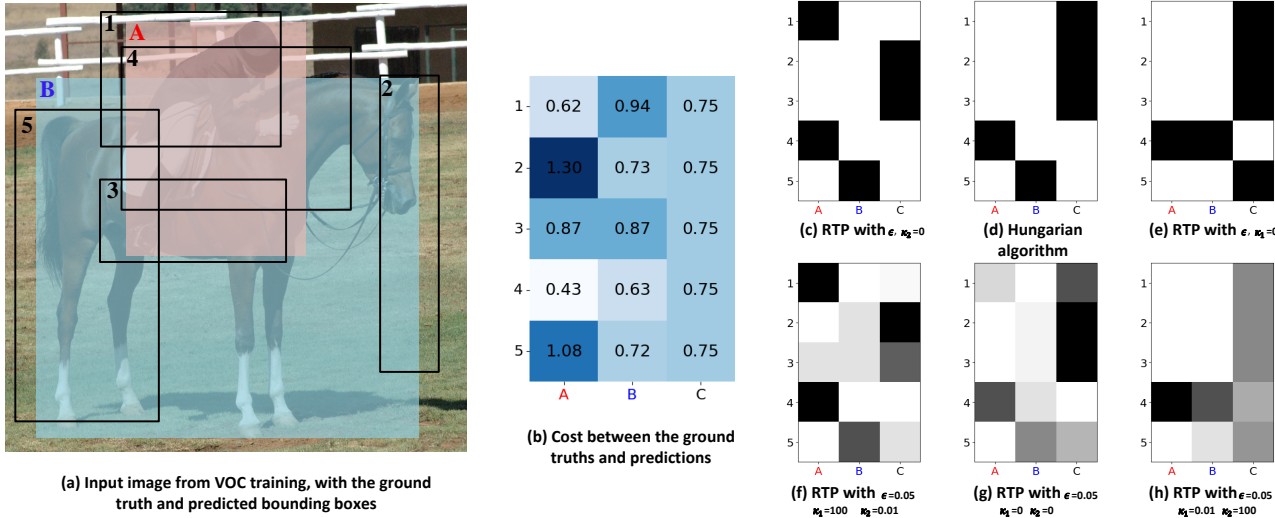

Figure 2: Matching analysis using the Hungarian algorithm and our model between ground-truth objects (A, B) and prediction boxes (1-5). We selected an input image with highly overlapped objects. Subfigures (c-h) illustrate the matching through color density, with black indicating a higher match and lighter colors (approaching white) signifying lower or no matches. This visualization helps in understanding the effectiveness of the matching process. See Section 2 for a detailed explanation.

mechanism in the decoder is the bottleneck to training efficacy, and suggested an encoder-only architecture as a solution. Gao et al.[10] aimed to streamline the cross-attention process by integrating a Gaussian before regulating attention within the model. Another direction to enhance DETR is to refine its matching strategy, which is more relevant to our work. This focus stems from the critical role of matching in object detection [8, 13], where accurately pairing each predicted object with its ground-truth during training (as illustrated in Figure 2). Deform-DETR [42] enhances the DETR's efficiency and performance, particularly when dealing with small objects or objects that require finer spatial resolution. This model uses Hungarian matching via a new optimization technique to find the best one-to-one correspondence between predicted and ground-truth objects during training. [13] proposed Salient-DETR, a salient score between detected and actual objects to ensure that only the most relevant objects are matched. Stable-DETR [8] reveals that DETR's slow convergence and performance issues stem from an unstable matching problem. It addresses this by introducing a stable matching technique that adjusts matching costs to prioritize positional metrics over semantic scores. DN-DETR [16] uses parallel decoders with shared weights to process multiple sets of noisy queries. These queries are crafted by introducing noise to the ground-truth object, aiming to improve the model's accuracy. Group DETR [4] enhances DETR by adding multiple decoder groups, each designed to handle specific subsets of object queries. Both DN-DETR and Group DETR use one-to-one matching strategy for every group of object queries to ensure accurate alignments. DINO [38] advances this by incorporating dynamic anchors and denoising training, achieving the first state-of-the-art performance on the COCO benchmark among DETR variants. Meanwhile, DETA [23] explores one-to-many assignment strategies by adopting additional decoders, which significantly increase computational demands, as

noted by [40]. MS-DERT [40] refines the Hungarian algorithm by implementing mixed supervision, combining one-to-one and one-to-many matching to boost the training efficiency without raising the cost during inference. Different from these approaches, our model seeks optimal alignment by computing transport plans between prediction and ground-truth sets via Sinkhorn's algorithm [6], to find the best correspondence.

## 3.2 Optimal Transport Alignment

Originally, optimal transport (OT) [24] tackles the problem of finding the most cost-effective way to align two sets of points (or distributions). It looks for an optimal coupling (transport plan) between distributions $\mu$ and $\nu$, representing it as a joint probability distribution. In other words, if we define $U(\mu, \nu)$ as the space of probability distributions over $R^d$ with marginals $\mu$ and $\nu$, the optimal transport is the coupling $\Gamma \in U(\mu, \nu)$, which minimizes the following quantity: $\min \langle \Gamma, C_{ij} \rangle$, where $C_{ij}$ is the cost of moving $i$ (from $\mu$) to $j$ (from $\nu$), respectively. The use of OT has gained popularity in generative modeling [30], adversarial training [35], and domain adaptation [3], and many other disciplines [5, 32, 34], since the introduction of Sinkhorn's algorithm [6]. Recent efforts in computer vision include matching predicted classes through Wasserstein distance [11], developing a specialized loss function for handling rotated bounding boxes [36], and introducing new metrics to evaluate model performance [22]. OT and the Hungarian algorithm, while both aimed at solving the assignment problem by finding the best matches between elements of two sets, differ significantly in their methodological execution. The Hungarian algorithm is designed for one-to-one matching, while, OT uses a probabilistic coupling, allowing the mass of a predicted object to be distributed across multiple ground-truths and vice versa, which is more flexible in capturing challenges in the assignments [3, 5, 7, 31].

# 4 METHOD

We first revisit important details on the Hungarian algorithm and OT, which will be useful to describe our proposed model.

## 4.1 Notations

For each image, we have a set of ground-truth objects $O^* = \{o_1^*, o_2^*, \ldots, o_N^*\}$, and a set of predicted objects $\hat{O} = \{\hat{o}_1, \hat{o}_2, \ldots, \hat{o}_M\}$, where $N$ is the number of ground-truth objects and $M$ is the number of predicted objects ($M \geq N$). Each ground-truth $o_i^*$ and predictions $\hat{o}_j$ is represented by a combination of class label and bounding box coordinates, denoted as $o_i^* = [c_i^*, b_i^*]$ and $\hat{o}_j = [\hat{c}_j, \hat{b}_j]$, respectively. Throughout the paper, vectors are denoted by lowercase letters, and matrices by uppercase. $1_N$ is $N$-dimensional vectors of ones, and $1_{M \times N}$ denotes an $M \times N$ matrix, each element of which is 1. The probability simplices $\Delta^N$ and $\Delta^M$, defined as $\Delta^N := \{u \in \mathbb{R}^N : \sum_i u_i = 1\}$ and $\Delta^M := \{v \in \mathbb{R}^M : \sum_j v_j = 1\}$, represent the sets of all possible weights for discrete measures across $N$ and $M$.

## 4.2 Hungarian matching algorithm

DETR uses the Hungarian algorithm [15] to establish an one-to-one pairing between predicted and actual objects by minimizing matching costs. The objective is to find an optimal permutation $\sigma$ of the $M$ predictions that minimizes the total matching cost:

$$\sigma^* = \operatorname*{argmin}_{\sigma} \sum_{i=1}^{N} C\left(O_i^*, \hat{O}_{\sigma(i)}\right), \qquad (1)$$

where $C(O^*, \hat{O})$ is the cost of matching the $i$-th ground-truth object to a prediction indexed by $\sigma(i)$. $C$ is a weighted sum of a classification loss and a localization loss (bounding boxes), defined as

$$C_{ij} = -\log(p_j(c_i)) + \lambda_{bbox} L_{bbox}(b_i^*, \hat{b}_j) + \lambda_{GIoU}(1 - \text{GIoU}(b_i^*, \hat{b}_j)), \qquad (2)$$

in which $p_j(c_i)$ represents the probability that the $j$-th prediction correctly classifies the $i$-th ground-truth. $L_{bbox}$ and GIoU measure the localization error between predicted and actual bounding boxes, with $\lambda$ parameters tuning the importance of each component. While the Hungarian algorithm ensures unique pairings to minimize the matching cost, it cannot always align with the complex realties of object detection, where multiple predictions correspond to a single object due to overlaps or visual ambiguities [4, 13, 40]. Moreover, this matching approach does not consider the overall distribution of predictions or ground-truths, which means it evaluates matches individually, ignoring the broader context of how all predictions and ground-truths should ideally be distributed or matched [13, 25].

## 4.3 Optimal Transport (OT)

In the quest to enhance DETR performance, we use OT [24], a mathematical formulation that has been widely applied in various alignment problems. OT aims to minimize the cost of transporting 'mass' from one distribution to another. In the context of object detection, OT can align a set of predicted objects to a set of ground-truth objects. Each 'mass' corresponds to an object, whether a prediction or a ground-truth. Consider the ground-truth objects $\{o_i^*\}_{i=1}^{N}$ and the predicted objects $\{\hat{o}_j\}_{j=1}^{M}$, with their respective distribution weights $\mu \in \Delta^N$ and $\nu \in \Delta^M$. Here, $\mu$ and $\nu$ denote

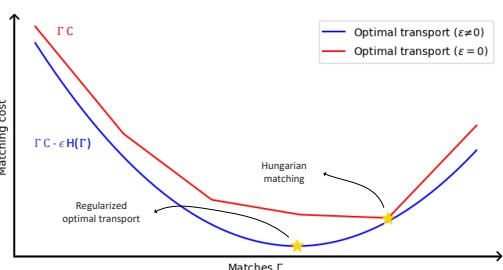

Figure 3: Effect of using regularization $H(\Gamma)$. The x-axis represents the matches $\Gamma$ (transport plan), which pairs predicted objects with ground-truth objects. The y-axis is the matching cost, with lower values denoting more cost-effective pairings. Each model's optimal match is highlighted with a star. The regularized transport plan (RTP) (blue line, with $\epsilon \neq 0$) shows reduced matching costs, indicating the benefits of the regularization term $H(\Gamma)$ in achieving a smooth distribution of matches. Interestingly, when $\epsilon = 0$, the model more closely resemble the one-to-one matches by the Hungarian algorithm.

the importance or confidence of each ground-truth and prediction, located at $o_i^*$ and $\hat{o}_j$, respectively. We then use a transportation cost $C \in \mathbb{R}^{M \times N}$ to compute a plan $\Gamma$ that minimizes the total cost of matching each prediction $j$ to a ground-truth $i$

$$\min_{\Gamma \in U(\mu, \nu)} \langle \Gamma, \mathbf{C} \rangle_F,$$

$$\textbf{where} \quad U(\mu, \nu) = \{\Gamma \in \mathbb{R}_+^{M \times N} : \Gamma 1_N = \mu, \quad \Gamma^T 1_M = \nu\}, \qquad (3)$$

$\langle \Gamma, \mathbf{C} \rangle_F$ is the Frobenius dot product between the transport plan and cost matrix. The cost $\mathbf{C} \geq 0$ represents how well (or poorly) a predicted object matches with a ground-truth. The polytope $U(\mu, \nu)$ is a set of transportation plans of dimension $M \times N$ to match predictions with ground truths, ensuring an optimal alignment under specified constraints. The cost matrix $\mathbf{C}$, with its elements $\mathbf{C}_{ij}$ is computed based on the Wasserstein distance (details are provided in the Suppl. file). Unlike OT, the Hungarian algorithm does not account for the distributions $\mu$ and $\nu$ of ground truths and predictions, treating all matches with equal importance [1, 7, 25, 40]. It enforces a strict one-to-one matching, leading to situations where some predictions may not match any ground-truth, thus deemed to match the background. OT, on the other hand, allows the distribution of each ground-truth object to be shared among several predictions. Similarly, a single prediction can correspond to multiple ground truths, providing a flexible matching process.

## 4.4 Regularized Transport Plan (RTP)

It is well-known that OT generates fully-dense transportation plans, meaning that every prediction is (fractionally) matched with all ground-truths [3, 5, 7]. This matching flexibility, however, introduces the problem of over-splitting. This occurs when the mass is spread too thinly across multiple predictions or ground truths, hence reducing the accuracy and reliability of the matching process. Indeed, OT alone cannot provide an effective alignment [29]. We incorporate a Kullback-Leibler (KL) divergence to relax the strict conservation of marginal constraints in OT by implementing

soft penalties [3]. This adjustment helps to control the distribution of probabilities, preventing over-splitting and ensuring that the matching process remains both accurate and fair. Given distributions $\mu \in \mathbb{R}^N$ for ground-truths and $\nu \in \mathbb{R}^M$ for predictions, along with a cost matrix $C$ where $C_{ij}$ reflects the cost of matching prediction $j$ to ground-truth $i$, our goal is to find a transportation plan $\Gamma \in \mathbb{R}^{M \times N}$ that minimizes the total transportation cost while ensuring that the resulting distribution of predictions and ground-truths (after transportation) closely aligns with their original distributions:

$$\min_{\Gamma \geq 0} \langle C, \Gamma \rangle + \kappa_1 D_{KL}(\Gamma \mathbf{1}_N \| \nu) + \kappa_2 D_{KL}(\Gamma^T \mathbf{1}_M \| \mu), \quad (4)$$

where $D_{\mathrm{KL}}$ is KL divergence. $\langle C, \Gamma \rangle = \sum_{i=1}^{N} \sum_{j=1}^{M} C_{ij} \Gamma_{ij}$ is the total transport cost, and $\kappa_1, \kappa_2$ are regularization parameters that balance the KL divergence terms. These terms align the transported mass distributions ($\Gamma^T \mathbf{1}_M$ and $\Gamma \mathbf{1}_N$) with the original distributions ($\nu$ and $\mu$), allowing the model to address discrepancies in the total distribution between predictions and ground-truths efficiently. The KL divergence measures the difference between the transported distribution ($\mathbf{1}_M^T, \mathbf{1}_N$) and the original distribution ($\mu, \nu$), as follow

$$\mathrm{KL}(\Gamma^T \mathbf{1}_M \| \mu) = \sum_{i=1}^{M} (\Gamma^T \mathbf{1}_M)_i \log(\frac{(\Gamma \mathbf{1}_M)_i}{\mu_i}), \quad (5)$$

for predictions, ensuring alignment with $\nu$, and

$$\mathrm{KL}(\Gamma \mathbf{1}_N \| \nu) = \sum_{j=1}^{N} (\Gamma \mathbf{1}_N)_j \log(\frac{(\Gamma^T \mathbf{1}_N)_j}{\nu_j}), \quad (6)$$

for ground truths, ensuring alignment with $\mu$. By embedding KL divergence, the finalized transport plan adheres closely to the original distributions of both predictions and ground-truths. This ensures that the matching process respects the inherent probabilistic nature of detections, accommodating scenarios where the number of predictions exceeds or falls short of the number of ground-truth.

*4.4.1 Mass Constraints with Entropy Regularization.* We now consider an entropic regularization of transport plan $\Gamma$ that controls the smoothness of the coupling and is particularly useful for reducing the computational complexity [3, 6, 29]. Computation of regularized OT relies on Sinkhorn's algorithm [6] which is notable for its efficiency and the ability to differentiate with respect to inputs. The resulting transport plan is easier to interpret because it provides a probabilistic view of the relationships between predicted and actual objects (Figure 2(f-h)). The regularized version of Eq. (4) is

$$\min_{\Gamma} \sum_{i=1}^{N} \sum_{j=1}^{M} C_{ij} \Gamma_{ij} + \kappa_1 \cdot \mathrm{KL}\left(T \mathbf{1}_N, \nu\right)$$
$$+ \kappa_2 \cdot \mathrm{KL}\left(T^T \mathbf{1}_M, \mu\right) - \epsilon H(\Gamma), \quad (7)$$

where $H(\Gamma) = -\sum_{ij} \Gamma_{ij} \log(\Gamma_{ij})$ is the entropy of the plan $\Gamma$ with $\epsilon$ as the regularization parameter. Sinkhorn's algorithm is well-suited for solving the regularized OT because it iteratively adjusts $\Gamma$ to satisfy the mass constraints while maximizing entropy, leading to a more balanced and flexible assignment [5] and this smoothing effect shown in Figure 3. However, reducing it to ($\epsilon \to 0$) and KL = 0, leads to numerical instability because the benefits of the sinkhorn algorithm and parallelization are no longer applicable.

---

**Algorithm 1** Regularized Transport Plan (RTP) Matching

---

**Require:** $C$: cost matrix ($M \times N$), $\mu$: distribution of ground-truths ($N$-vector), $\nu$: distribution of predictions ($M$-vector), $\kappa$: regularization parameter for KL divergence, $\epsilon$: regularization parameter for entropy

**Ensure:** $\Gamma$: Optimal Assignment ($M \times N$ matrix)
1: $K[i, j] \leftarrow \exp(-C[i, j]/\epsilon)$                    ▷ Initialize Gram matrix
2: $u \leftarrow \mathbf{1}_N,\ v \leftarrow \mathbf{1}_M$          ▷ Initialize scaling vectors
3: **while** not converged **do**                              ▷ Sinkhorn algorithm
4:     $u[i] \leftarrow \mu[i]/(Kv)[i]$              ▷ Update scaling vector $u \in \mu$
5:     $v[j] \leftarrow \nu[j]/(K^\top u)[j]$          ▷ Update scaling vector $v \in \nu$
6: **end while**
7: $\Gamma[i, j] \leftarrow u[i] \cdot K[i, j] \cdot v[j]$           ▷ Compute transport plan
8: **for** $i \leftarrow 1$ to $N$ **do**       ▷ Adjust $\Gamma$ for KL divergence w.r.t. $\mu$
9:     rowSum $\leftarrow \sum_j \Gamma[i, j]$
10:     **if** rowSum $\neq \mu[i]$ **then**
11:         $\Gamma[i, :]$ to make the row sums align with $\mu[i]$.
12:     **end if**
13: **end for**
14: **for** $j \leftarrow 1$ to $M$ **do**       ▷ Adjust $\Gamma$ for KL divergence w.r.t. $\nu$
15:     colSum $\leftarrow \sum_i \Gamma[i, j]$
16:     **if** colSum $\neq \nu[j]$ **then**
17:         $\Gamma[:, j]$ to make the column sums align with $\nu[j]$.
18:     **end if**
19: **end for**
20: **return** $\Gamma$

---

In this case, we recover the exact OT (Eq. (3)) which has shown one-to-one mapping as discussed in [5] and shown in Figure 3.

## 4.5 Training

In classical DETR, predictions are matched to ground-truth bounding boxes using the Hungarian algorithm, through a cost function $C_{\mathrm{match}}(\hat{o}_j, o_i^*) = \lambda_{\mathrm{prob}}(1 - \langle \hat{c}_j, c_i^* \rangle) + \lambda_l \|\hat{b}_j - b_i^*\| + \lambda_{GIoU}(1 - GIoU(\hat{b}_j, b_i^*))$ in Eq. (2). This approach requires an equal number of predictions and ground-truth boxes for effective matching. However, our procedure is as follows: for each image, generate predictions ($\hat{c}_j, \hat{b}_j$) and compare these to the ground-truths ($c_i^*, b_i^*$) using the cost function $C$. The cost matrix integrates both classification and localization losses and defined as: $C(\hat{o}_j, o_i^*) = L_{\mathrm{cls}}(\hat{c}_j, c_i^*) + L_{\mathrm{loc}}(\hat{b}_j, b_i^*)$. The first component ensures that similar object classes are matched together by evaluating the probability that the $j$-th prediction correctly classifies the $i$-th ground-truth. The second component addresses the spatial alignment by measuring the discrepancy between the predicted bounding box and the ground-truth bounding box. To further enhance the matching process, we incorporate entropic regularization into the cost function (Eq. (7)) and employ Sinkhorn's algorithm to compute the optimal transport plan $\Gamma$. The proposed matching strategy is given in Algorithm 1. We also train the model by cross-entropy (CE) loss, $\sum_{i=1}^{N} \sum_{j=1}^{M} \Gamma_{ij} \cdot L_{\mathrm{train}}(\hat{o}_j, o_i^*)$, where $L_{\mathrm{train}}(\hat{o}_j, o_i^*) = \lambda_{\mathrm{CrossE}}(\hat{c}_j, c_i^*) + \lambda_l \|\hat{b}_j - b_i^*\| + \lambda_{GIoU}(1 - GIoU(\hat{b}_j, b_i^*))$. Sinkhorn's algorithm effectively guides each prediction $\hat{o}_j$ towards its best-matching ground truth $o_j^*$ based on the calculated weights $\Gamma_{ij}$. This can provide an accurate matching between predictions and ground-truth objects.

| Method | Backbone | #epochs | AP ↑ | AP$_{50}$ ↑ | AP$_{75}$ ↑ | AP$_S$ ↑ | AP$_M$ ↑ | AP$_L$ ↑ |
|---|---|---|---|---|---|---|---|---|
| Deform-DETR [42] | ResN50 | 50 | 46.9 | 65.6 | 51.0 | 29.6 | 50.1 | 61.6 |
| Sparse-DETR [28] | ResN50 | 50 | 46.3 | 66.0 | 50.1 | 29.0 | 49.5 | 60.8 |
| Effcient-DETR [37] | ResN50 | 36 | 45.1 | 63.1 | 49.1 | 28.3 | 48.4 | 59.0 |
| H-DETR [14] | ResN50 | 36 | 50.0 | 68.3 | 54.4 | 32.9 | 52.7 | 65.3 |
| DN-DETR [16] | ResN50 | 12 | 43.4 | 61.9 | 47.2 | 24.8 | 46.8 | 59.4 |
| Rank-DRT [25] | ResN50 | 12 | 50.2 | 67.7 | 55.0 | 34.1 | 53.6 | 64.0 |
| DINO-DETR [38] | ResN50 | 12 | 49.0 | 66.6 | 53.5 | 32.0 | 52.3 | 63.0 |
| Salience-DETR [13] | ResN50 | 12 | 49.4 | 67.1 | 53.8 | 32.7 | 53.0 | 63.1 |
| H-DETR [14] | ResN50 | 12 | 48.7 | 66.4 | 52.9 | 31.2 | 51.5 | 63.5 |
| RTP-DETR | ResN50 | 12 | 50.7 | 67.9 | 55.2 | 34.7 | 53.8 | 64.2 |

**Table 1: Comparison of our approach (RTP-DETR) with top-performing DETR-based models using the ResNet50 backbone on the COCO dataset. Our model consistently surpasses or performs competitively with respect to the state-of-the-art baselines.**

| Method | Matching | Backbone | #epochs | AP ↑ | AP$_{50}$ ↑ | AP$_{75}$ ↑ | AP$_S$ ↑ | AP$_M$ ↑ | AP$_L$ ↑ |
|---|---|---|---|---|---|---|---|---|---|
| DINO-DETR [38] | | ResN50 | 36 | 50.9 | 69.0 | 55.3 | 34.6 | 54.1 | 64.6 |
| DINO-DETR [38] | Baseline | ResN50 | 12 | 49.0 | 66.6 | 53.5 | 32.0 | 52.3 | 63.0 |
| | Rank [25] | ResN50 | 12 | 50.4 | 67.9 | 55.2 | 33.6 | 53.8 | 64.2 |
| | Stable-DINO [19] | ResN50 | 12 | 50.4 | 67.4 | 55.0 | 32.9 | 54.0 | 65.5 |
| | RTP | ResN50 | 12 | 50.6 | 68.2 | 55.2 | 33.5 | 54.0 | 65.5 |
| DINO-DETR [38] | | Swin-L | 36 | 58.0 | 77.1 | 66.3 | 41.3 | 62.1 | 73.6 |
| DINO-DETR [38] | Baseline | Swin-L | 12 | 56.8 | 75.4 | 62.3 | 40.0 | 60.5 | 73.2 |
| | Rank [25] | Swin-L | 12 | 57.5 | 76.0 | 63.4 | 41.6 | 61.4 | 73.8 |
| | Stable-DINO [19] | Swin-L | 12 | 57.7 | 75.7 | 63.4 | 39.8 | 62.0 | 74.7 |
| | RTP | Swin-L | 12 | 57.9 | 76.1 | 63.6 | 41.5 | 62.3 | 74.9 |

**Table 2: Enhancing object detection performance with DINO-DETR on the COCO val2017 dataset, using ResNet50 [12] and Swin-Large [21] backbones. RTP-DETR is presented as a complementary model to existing methods and achieves consistent enhancements in performance. We also include the result of DINO-DETR trained for 36 epochs as a reference for comparison.**

## 5 EXPERIMENTAL RESULTS

### 5.1 Training settings

Our evaluation was conducted on two widely recognized datasets to validate the effectiveness of our matching strategy, which has a time complexity of $O(NM \log(NM))$, where $N$ is the number of ground truth objects and $M$ is the number of predicted objects. The first dataset, COCO object detection [17], includes 118,287 training images and a validation set of 5,000 images. The second dataset, PASCAL-VOC object detection [9], is fine-tuned on VOC train-val07+12 with approximately 16.5k images and evaluated on the test2007 set. We use mean Average Precision (AP) and mean Average Recall (AR) as primary metrics for evaluation. We compare the performance of our model (RTP-DETR) against several DETR variants, including DN-DETR [16], Salience-DETR [13], Rank-DETR [25], DINO [38], and Group-DETR [4]. All models are trained using the same set of hyperparameters: $\lambda_{prob} = \lambda_{CrossE} = 2$, $\lambda_{GIoU} = 2$ and $\lambda_l = 5$. Regarding our model-specific parameters, the entropic regularization is set to $\epsilon = \epsilon_0/(\log(2M) + 1)$ with $\epsilon_0 = 0.17$. This adjusts the entropic regularization based on the number of predictions, enhancing the stability and convergence of the model. We used multiple values of $\kappa_2$ but $\kappa_1$ is fixed to a large value $\kappa_1 = 100$.

### 5.2 Comparison with DETR-based methods

Table 1 shows the performance of RTP-DETR compared with other high-performing DETR-based methods on the COCO object detection val2017 set using ResNet-20 [12]. With only 12 training epochs, RTP-DETR achieves an impressive AP of 50.7%, which suppresses H-DETR by +2.0% and exceeds the most recent state-of-the-art Salience-DETR and Rank-DETR, by +1.3% and +0.5%. Interestingly, we observe notable improvements in AP$_{75}$, demonstrating the advantage of our approach at higher thresholds. Additionally, Table 2 illustrates the effectiveness of our model in enhancing DINO-DETR, a training-efficient DETR variant that has received significant attention in the object detection task. The results are obtained using two different backbones, ResNet-50 [12] and Swin-Large [21]. We also compare our model with other integrations such as Rank-DINO and Stable-DINO. The improvements with our model are notable: there's an increase of 1.6% in AP performance when using the ResNet-50 backbone (49% vs. 50.6%) and 1.1% with Swin-L (56.8% vs. 59.7%). At a higher IoU threshold (AP$_{75}$), the enhancements become even more pronounced, achieving +1.7% improvement with ResNet-50 and +1.3% with Swin-L. These results indicate our model's robustness and generalizability across various DETR-based models.

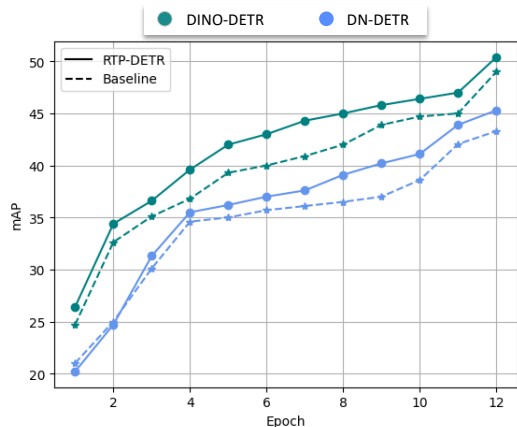

**Figure 4: Convergence curves. RTP accelerates the training process for different variants of DETR. The baseline models and our RTP counterparts are shown by dotted and solid lines, respectively. The horizontal axis denotes the number of epochs, the vertical axis is the AP evaluated on COCO.**

### 5.3 Combination with DETR variants

Table 3 summarizes the integration of our RTP with several DETR variations. As can be seen, when combined with Deform-DETR, the AP has increased by +1.8, from 46.9% to 48.7%. Similarly, the performance of Group-DETR has improved by +0.7, moving from 48% to 48.7%, while using only 12 epochs schedule. Additionally, our RTP has enhanced DINO-DETR's performance, boosting its AP by +1.4 over a 24-epoch and by an additional +1.6 AP when the training is extended to 36 epochs. It's important to note that many of these methods rely on a one-to-one matching strategy, with modifications mainly focused on optimizing the Hungarian algorithm to better accommodate one-to-many relationships between predictions and ground truths during training. Unlike these advances, RTP utilizes a different matching strategy, providing additional advantages of regularization that contribute to the performance gains observed.

### 5.4 Convergence

Adding entropy regularization into the transport plan (Eq. (7)), yields an efficient and seamless matching process. Figure 3 provides a clear illustration of this improvement by showing the effectiveness and speed of the RTP matching process. Furthermore, Figure 1 details the convergence behavior of our model, demonstrating that RTP enables significantly faster convergence compared to standard DETR. The slower convergence rates of conventional DETR can be attributed to the discrete and unstable nature of the Hungarian algorithm, particularly during the early stages of training [19, 25, 42]. We also conducted a comparative analysis of our model alongside two prominent DETR variants, DINO [38] and DN [16], known for their effective one-to-many matching capabilities. The models use a ResNet-50 backbone and are trained over 12 epochs, as shown in Figure 4. This comparison demonstrates that RTP not only refines the matching process but also significantly accelerates model convergence during training, thereby providing a robust solution to the inherent limitations of the Hungarian matching algorithm.

| Method | Matching | # epochs | AP | AR |
|---|---|---|---|---|
| Deform-DETR | - | 50 | 46.9 | 57.3 |
| Deform-DETR | RTP | 50 | 48.7 (+1.8) | 58.6 (+1.3) |
| DINO-DETR | - | 24 | 50.4 | 65.4 |
| DINO-DETR | RTP | 24 | 51.8 (+1.4) | 66.3 (+0.9) |
| Group-DERTR | - | 12 | 48.0 | 67.2 |
| Group-DERTR | RTP | 12 | 48.7 (+0.7) | 67.9 (+0.7) |
| Salience-DETR | - | 12 | 49.2 | 63.5 |
| Salience-DETR | RTP | 12 | 49.8 (+0.6) | 64.0 (+0.5) |
| Rank-DETR | - | 12 | 50.2 | 67.9 |
| Rank-DETR | RTP | 12 | 50.6 (+0.4) | 68.2 (+0.3) |

**Table 3: Combination with other methods. RTP is a complementary approach that consistently improves performance.**

| Method | # epochs | AP | AR |
|---|---|---|---|
| Align-DETR [1] | 12 | 50.2 | 61.7 |
| + RTP | 12 | 50.6 (+0.4) | 62.3 (+0.6) |
| Align-DETR [1] | 24 | 51.3 | 62.4 |
| + RTP | 24 | 51.9 (+0.7) | 62.9 (+0.5) |

**Table 4: The effect of combining RTP with Align-DETR demonstrates a significant improvement in performance.**

### 5.5 Ablation

*5.5.1 Computation Time.* Figure 5 compares the time complexity of different matching models, focusing on the Hungarian algorithm, RTP without KL divergence ($\kappa = 0$), and the full implementation of RTP. We observe that both RTP variants are relatively stable, even with increasing numbers of predictions. This consistency is due to the use of entropy regularization ($H(\Gamma)$) within the OT framework, which effectively controls computational complexity and ensures only minimal increases in processing time as predictions scale. On the other hand, the Hungarian algorithm shows a noticeable increase in computation time as the number of predictions grows (beyond 900 predictions). This behavior points to a limitation of the Hungarian algorithm when dealing with high-volume predictions [1, 19]. The performance of our models is particularly notable when handling large numbers of predictions. Through GPU parallelization of the Sinkhorn algorithm, we achieve more than 30× speedup. These computations were performed on an Nvidia TITAN X GPU. We also used SSD300 which makes 8, 732 predictions, as a reference point. The implementation of SSD is provided in the Suppl. file.

*5.5.2 Adoption of IoU-Optimized Loss.* We explore the combination of RTP with recent DETR variants, particularly those leveraging IoU-optimized loss to enhance DETR performance significantly [1, 19, 25]. By integrating RTP with Align-DETR [1], we achieve complementary effects between RTP and IoU-aware loss. As illustrated in Table 4, this combination results in an increase of +0.4 AP for a 12-epoch training schedule and +0.7 for a 24-epoch schedule.

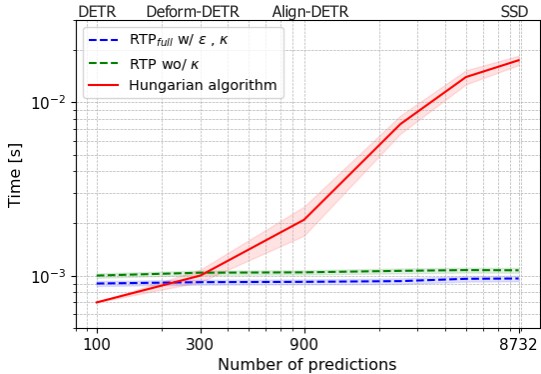

Figure 5: Computation time and its standard deviation on the COCO using various matching techniques with a batch of size 16. RTP ($\kappa = 0$): represents our model implemented with entropic regularization but without KL divergence terms, highlighting the direct influence of entropic regularization on the training process. RTP$_{full}$: integrates both entropic regularization and KL divergence providing a comprehensive view of the regularization effect on on training efficiency.

*5.5.3 Influence of Hyperparameters.* We illustrate the influence of two hyperparameters, $\epsilon_0$ and $\kappa_2$, in our matching technique. As defined in (5.1), the regularization parameter $\epsilon$ is set to $\epsilon_0/(\log(2M) + 1)$, which scales based on the number of predictions $M$. As $M$ increases, $\log(2M)$ enlarges, causing $\epsilon$ to decrease. Figure 7 details the impact of the regularization $\epsilon_0$ on the matching process. When $\epsilon_0$ is a very small value, the effect of $\epsilon$ becomes minimal, and at $\epsilon_0 = 0$, entropy regularization no longer plays a role in the matching process. Conversely, larger $\epsilon_0$ values increase the corresponding $\epsilon$, potentially risking overfitting or loss of essential details in detection. Empirical results indicate that our model performs best when $\epsilon_0$ is within the range $[0.15 - 0.25]$. Additionally, our analysis on the hyperparameter $\kappa_2$ is presented in Figure 7. Our model achieves optimal results when $\kappa_2 = 0.01$. We maintain $\kappa_1 = 100$, as changing this value leads to an object imbalance problem, emphasizing its importance in preserving the accuracy of ground-truth alignment. Indeed, $\kappa_1$ ensures accurate alignment of ground-truths to predictions, while $\kappa_2$ allows flexibility through probabilistic matching.

*5.5.4 Visualization.* Two different configurations of our proposed model are visualized in Figure 6. The corresponding attention maps for each image are presented on the left, with areas of intense red indicating the predicted object locations. The top row presents the result of the model without regularization term (RTP-DETR w/ $\epsilon = 0$), representing a simpler version of the model. The bottom row displays our full implementation of the model. The distinct difference in attention map concentration between the rows is evident, particularly in the last image where the soccer field is populated with multiple small objects. The basic model struggles to handle multiple objects, resulting in scattered and unclear attention maps. In contrast, the optimized model (bottom row) exhibits well-defined and focused attention, accurately capturing the shape and position of each object, even in densely packed scenes. This demonstrates

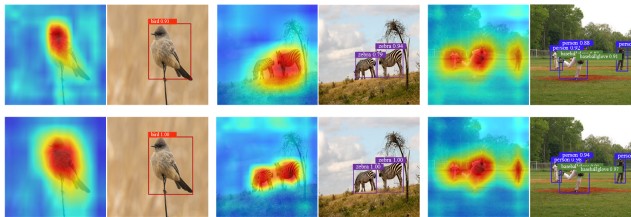
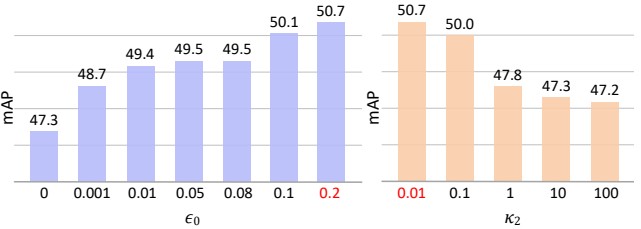

Figure 6: Visualization on sample images from COCO test set. The left side of each image displays the respective attention maps generated. The top row shows results from RTP-DETR without the regularization term, whereas the bottom row shows our full implementation. The attention maps in the second (bottom) row demonstrate that the regularized model more accurately identifies and delineates object shapes and positions, particularly in complex and densely packed scenes.

Figure 7: Influence of the hyperparameters $\epsilon$ and $\kappa_2$. The x-axis displays $\epsilon_0$ on the left and $\kappa_2$ on the right, while the y-axis shows AP evaluated on COCO. The best-performing values for each hyperparameter are highlighted in red.

that regularization helps the model to better concentrate on relevant parts of the image, thereby improving object detection accuracy.

# 6 CONCLUSION

In this paper, we have examined the comparative effectiveness of the Hungarian algorithm and transportation plan in object detection, focusing on reducing the matching costs between predicted and actual data. The Hungarian algorithm, with its strict one-to-one matching, is effective when the number of predictions matches the number of ground truths exactly. However, this rigid strategy limits its application and fails to capture the complexities and variations present in real-world data. In contrast, regularized optimal transport, through a probabilistic coupling, offers a flexible solution that accounts for the entire distribution of matches. This strategy not only facilitates a more comprehensive understanding of the *relationships* between predictions and ground-truths objects but also handles the discrepancies in set sizes. Our findings demonstrate that the transportation plan with entropic regularization consistently outperforms the Hungarian method by providing a more accurate and flexible alignment without relying on predefined thresholds. In future, we extend the regularized OT framework to zero-shot detection scenarios, where the model must detect objects not seen during training. This requires developing transferable transport plans that can generalize across different object categories and domains.

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
