# OpenReview forum: "Fractional Correspondence Framework in Detection Transformer"
_acmmm.org/ACMMM/2024/Conference — MM2024 Poster_

### Official Review · Reviewer_WfU8 · 2024-05-13

**Rating:** 3
**Confidence:** 2

**Summary:**

This paper proposes the Regularized Transport Plan (RTP) for object detection. The RTP is computed using the differentiable Sinkhorn algorithm to allow for soft, fractional matching rather than strict one-to-one assignments. This approach enhances DETR adaptability to complex object detection scenarios, providing a nuanced and precise assessment of disparities between prediction and ground-truth distributions. Evaluations on the MS-COCO benchmarks demonstrate the effectiveness of the approach.

**Strengths:**

1. RTP can make DETR converge faster and improve detection performance.
2. RTP have a lower computational cost than Hungarian algorithm when the number of prediction is larger than 300.
3. RTP can be combined with many variants of DETR for further improvements.

**Limitations:**

1. The paper only shows evaluation of 2D object detection on COCO dataset. How about the performance on 2D panoptic segmentation, 2D pose estimation, 3D object detection, and multi-object tracking?
2. More explanation on why RTP improve the matching. Like visualization of features and comparing results with DETR.
3. The paper focuses on improving a component of DETR. I think the paper is suit for computer vision and machine learning conferences.

**Suitability:**

1

---

### Official Review · Reviewer_3bWv · 2024-05-22

**Rating:** 4
**Confidence:** 2

**Summary:**

This paper proposes a regularized transport plan to find the best alignment between predictions and ground-truths. It enhances DETR adaptability to complex object detection scenarios.

**Strengths:**

The method is formulated in detail and clearly.

Extensive experiments are conducted to prove its effectiveness.

**Limitations:**

The paper claims that DETR is unsuitable for complex scenarios, such as detecting small or dense objects. Concrete results are needed to demonstrate the effectiveness of RTP in these scenarios.

Some parts of the writing and organization need improvement. For example, in Section 2, the significance of the hyperparameters (what large and small values represent) is not clearly explained in advance.

**Suitability:**

2

---

### Official Review · Reviewer_rTGN · 2024-05-24

**Rating:** 4
**Confidence:** 3

**Summary:**

This paper is motivated by the fact that one-to-one Hungarian Matching does not effectively model the varying densities and distributions
 of objects in DETRs. The author propose a new matching algorithm Regularized Transport Plan (RTP) to handle this problem.
Experiment results on COCO show it advantages by overpass the state-of-the-art DETRs (Rank-DETR, Stable-DINO).

**Strengths:**

1. The ground truth - model prediction problem is modeled as a optimal transport problem in this paper, which is novel in the DETR research area.
2. This paper is well motivation and find a very interesting research question. The authors thinks the matching procedure from a new perspective.
3. The experiment results is very solid, validating its effectiveness.

**Limitations:**

1. Figure 1 is not very fair. The original DETR is very naive, and compare the proposed method (with contains many advances in previous works) with it is not fair. The results on VOC is also not persuasive, it should be replaced with some large dataset.
2. The reason for why the proposed method leading to a more balanced and flexible assignment  is not very well clarified. Please detailed discuss it.
3.Minors:
1) spell error (e.g. DET --> DETR in L588)
2) spell error (e.g. DERTR --> DETR in L762)
3) the font  size in images should match that in the paper.

**Suitability:**

3

---

### Official Review · Reviewer_CtCw · 2024-05-25

**Rating:** 5
**Confidence:** 2

**Summary:**

This paper analyzes the drawbacks of the matching strategy in DETR-based detection methods. It proposes a regularized transport method for the alignment between predictions and ground truths, which accelerates convergence during the training process. In experiments, this paper compares the proposed method with several DETR-based detection methods and achieves competitive results. Additionally, as a plug-and-play matching strategy, it has been combined and compared with multiple baselines, resulting in consistent performance improvements.

**Strengths:**

1. The matching strategy in DETR-based detection methods has been deeply analyzed.
2. The paper's writing and algorithm descriptions are clear and easy to read.
3. This paper offers a novel and unique perspective and approach to DETR-based detection methods.

**Limitations:**

1. I think that a time complexity analysis for Algorithm 1 is very necessary, not just a comparison of average computation times.
2. In the visualization in Figure 6, the authors mention that “the regularized model appears to be better at discerning the exact shape and position of the objects.” Is this enhancement in the model’s ability to perceive the shape of the objects due to smoother and faster convergence? I hope the authors can provide a more detailed explanation to help readers understand.
3. Using color density to compare the degree of matching in Figure 2 is very clear, but I think it would be more intuitive to compare the Hungarian matching and the proposed matching strategy based on visual results in the images. For example, compare the different prediction boxes matched to the same target in the ground truth under the two matching strategies, and explain the drawbacks of the Hungarian matching method and the advantages of the proposed method.
4. According to Table 3 and Table 4, RTP does not show much performance improvement on object detection datasets. On the other hand, based on Figure 3 and Figure 4, the model’s convergence speed and results improve after using RTP. I wonder if the authors could analyze their thoughts on balancing these two aspects, i.e., the significance of better convergence and minor performance gains in this field?

**Suitability:**

2

---

### Meta-Review · Area_Chair_xeNP · 2024-07-07

**Recommendation:** Accept (Poster)
**Confidence:** 5

**Metareview:**

The paper introduces the Regularized Transport Plan (RTP) for object detection, enhancing DETR adaptability in complex scenarios, as demonstrated on the MS-COCO benchmarks. Initial reviews were mixed, with concerns about the paper's focus and clarity. Post-rebuttal, reviewers acknowledged improvements made by the authors in addressing these concerns, leading to an improved average rating leaning towards acceptance.

The recommendation is to accept the paper for publication, as the paper presents a novel and meaningful contribution to the field of object detection.

---

### Meta-Review · Senior_Area_Chairs · 2024-07-10

**Recommendation:** Accept (Poster)
**Confidence:** 4

**Metareview:**

All the reviewers gave positive ratings and tend to accept the paper. SAC and AC agree with the reviewers and recommend acceptance of the paper.